

# In situ observation of atmospheric oxygen and carbon dioxide in the North Pacific using a cargo ship

Yu Hoshina[1], Yasunori Tohjima[1], Keiichi Katsumata[1], Toshinobu Machida[1], and Shin-ichiro Nakaoka[1]

[1]National Institute for Environmental Studies, Tsukuba, 305-8506, Japan

*Correspondence to*: Yu Hoshina (hoshina.yu@nies.go.jp)

**Abstract.** Atmospheric oxygen ($O_2$) and carbon dioxide ($CO_2$) mixing ratios in the North Pacific were observed onboard a cargo ship, the *New Century 2* (NC2), while it cruised between Japan and the United States between December 2015 and November 2016. A fuel cell analyzer and a non-dispersive infrared analyzer were used for measurement of $O_2$ and $CO_2$, respectively. To achieve parts-per-million precision for the $O_2$ measurements, we
precisely controlled the flow rates of the sample and reference air introduced into the analyzers and the outlet pressure. A relatively low airflow rate ($10 \ cm^3 \ min^{-1}$) was adopted to reduce the consumption rate of the reference gases. In the laboratory, the system achieved measurement precisions of 3.8 per meg for the $O_2/N_2$ ratio, which is commonly used to express atmospheric $O_2$ variation, and 0.1 ppm for the $CO_2$ mixing ratio. After the in situ observation started onboard NC2, we found that the ship's motion caused false wavy variations of the $O_2$ signal
with an amplitude of more than several tens of ppm and a period of about 20 s. Although we have not resolved the problem at this stage, hourly averaging considerably suppressed the variation associated with ship motion. Comparison between the in situ observation and flask sampling of air samples onboard NC2 showed that the averaged differences (in situ – flask) and the standard deviations ($\pm 1\sigma$) are $-2.1\pm9.2$ per meg for the $O_2/N_2$ ratio and $-0.02\pm0.33$ ppm for the $CO_2$ mixing ratio. We compared the 1 yr of in situ data for atmospheric potential
oxygen (APO) ($= O_2 + 1.1 \times CO_2$) obtained from the broad middle-latitudes region (140°E–130°W, 29°N–45°N) with previous flask sampling data from the North Pacific. This comparison showed that longitudinal differences in the seasonal amplitude of APO, ranging from 51 to 73 per meg, were smaller than the latitudinal differences.

## 1 Introduction

The balance of $CO_2$ emissions from fossil fuel combustion, land biotic $CO_2$ uptake, and ocean $CO_2$ uptake
determines long-term change in the atmospheric $CO_2$ burden. At the same time, fossil fuel combustion consumes atmospheric $O_2$ while the land biotic $CO_2$ uptake is accompanied by emission of $O_2$ into the atmosphere. Additionally, today's ocean is considered to be a weak source of $O_2$ because of recent ocean warming (Bopp et al.,





2002; Keeling and Garcia, 2002; Plattner et al., 2002). The $CO_2$ and $O_2$ exchanges for biotic processes and fossil fuel combustion are stoichiometrically related, and the fossil fuel consumption rate can be reliably estimated from energy statistics. Therefore, coupled measurements of the atmospheric $CO_2$ and $O_2$ have been used to constrain the global $CO_2$ budgets by simultaneously solving the equations for the atmospheric $CO_2$ and $O_2$ budgets (Keeling and
Shertz, 1992; Battle et al., 2000, 2006; Manning and Keeling, 2006; Tohjima et al., 2008; Ishidoya et al., 2012).

Atmospheric $O_2$ measurements are also useful for understanding air-sea gas exchange. This is based on the fact that the air-sea exchange of $O_2$ is more than one order of magnitude faster than that of $CO_2$; chemical equilibrium of the dissolved inorganic carbons (dissolved $CO_2$, bicarbonate and carbonate ions) in seawater suppresses the air-sea exchange of $CO_2$ (e.g., Keeling et al., 1993). For example, seasonality in the ocean biotic activity and the sea
surface temperature is poorly reflected in air-sea $CO_2$ exchange, while oceanic $O_2$ fluxes show clear seasonal variations. The role of atmospheric $O_2$ as a tracer of air-sea gas exchange is emphasized by the introduction of a tracer of atmospheric potential oxygen (APO), which is defined as APO = $O_2$ + 1.1 × $CO_2$ (Stephens et al., 1998), where 1.1 represents the $-O_2/CO_2$ exchange ratio associated with land biotic activity (Severinghaus, 1995). Since the APO is invariant with respect to the land biotic $O_2$ and $CO_2$ exchange and predominantly reflects the air-sea gas
exchange, spatiotemporal variations in APO have been used to study gas fluxes associated with ocean ventilation (Lueker et al., 2003) and ocean spring bloom (Yamagishi et al., 2008) to estimate net ocean production (Keeling and Shertz, 1992; Balkanski et al., 1999; Nevison et al., 2012) and to validate ocean biogeochemical models (Stephens et al., 1998; Naegler et al., 2007: Nevison et al., 2008, 2015, 2016).

The precision required to detect atmospheric $O_2$ variation is at the μmol/mol (ppm) level, which is considerably
smaller than the atmospheric $O_2$ mole fraction of about 21%. First, Keeling (1988) developed an atmospheric $O_2$ measurement technique based on an interferometer and showed the usefulness of $O_2$ measurements to study the global carbon cycle. Since then, several $O_2$ measurement techniques based on a mass spectrometer (Bender et al., 1994), a paramagnetic analyzer (Manning et al., 1999), a fuel cell analyzer (Stephens et al., 2007), and a vacuum ultraviolet absorption photometer (Stephens et al., 2003) have been developed and applied to atmospheric
observations (cf. Keeling and Manning, 2014). When the change in atmospheric $O_2$ concentration is compared with that of $CO_2$, it is expressed as a deviation of the $O_2/N_2$ ratio from an arbitrary reference according to

$$\delta(O_2/N_2) = \frac{(O_2/N_2)_{sam}}{(O_2/N_2)_{ref}} - 1 \qquad (1)$$



where subscripts *sam* and *ref* represent sample and reference gases, respectively, and the $\delta(O_2/N_2)$ value is usually converted to a "per meg" value, which approximates parts per million, by multiplying it by $10^6$ (Keeling and Shurtz, 1992). The reason for not using the mole fraction for $O_2$ is that, for example, a change of 1 μmol of $O_2$ per mol of dry air does not necessarily result in a 1-ppm change in the $O_2$ mole fraction but always corresponds to a 4.77 per

meg change in the $\delta(O_2/N_2)$ value.

The National Institute of Environmental Studies, Japan (NIES) also developed a technique to measure the atmospheric $O_2/N_2$ ratio based on a gas chromatograph equipped with a thermal conductivity detector (GC/TCD) (Tohjima, 2000). NIES began measuring atmospheric $O_2/N_2$ and $CO_2$ by collecting air samples in glass flasks at two monitoring stations, Hateruma Island in July 1997 and Cape Ochi-ishi in December 1998 (Tohjima et al., 2003,

2008). Additionally, to extend the observation area, we started flask sampling onboard cargo ships sailing between Japan and Australia/New Zealand (Oceanian route) and between Japan and North America (North American route) in December 2011 (Tohjima et al., 2005, 2012). Moreover, in situ measurements of the atmospheric $O_2/N_2$ ratio using the GC/TCD technique also started onboard a cargo ship between Japan and Australia/New Zealand in 2007 (Yamagishi et al., 2012).

These $O_2$ and $CO_2$ data from widespread Pacific regions were used to investigate the spatial distribution of the climatological seasonal cycle of APO and the annual mean values of APO (Tohjima et al., 2012). Latitudinal transection of the data in the western Pacific region revealed that the magnitude of equatorial elevation appearing in the annual mean APO was associated with the El Niño–Southern Oscillation cycle (Tohjima et al., 2015). The relatively high spatiotemporal sampling density allowed us to investigate the interannual variation of the annual

mean APO in the western Pacific. In contrast, the spatiotemporally sporadic APO data obtained from the North American route make it difficult to investigate interannual variations in the northern and eastern North Pacific. In 2015, Pickers et al. (2017) started in situ observations of atmospheric $O_2$ and $CO_2$ with a fuel cell analyzer and non-dispersive infrared analyzer onboard a commercial container ship regularly traveling in the Atlantic Ocean between Hamburg, Germany, and Buenos Aires, Argentina. They also showed the usefulness of continuous observation to

reveal the spatiotemporal APO distribution. Therefore, in situ measurement onboard a cargo ship on the North American route in the Pacific was also desired.

Since June 2014, atmospheric greenhouse gas measurements, including flask sampling (7 flasks per round trip), have been conducted on a cargo ship, the *New Century 2* (NC2) along the North Pacific route. We also had an opportunity to install an atmospheric $O_2$ measurement system onboard NC2. However, since the onboard space

allotted to us was limited, we had to make the measurement system smaller by reducing the number of cylinders



required for the system. In addition, since the processes for loading and unloading high-pressure gas cylinders on ocean-going ships are cumbersome, we needed to reduce the consumption rate of reference gases to reduce the cylinder exchange frequency. Considering these conditions, the GC/TCD technique, which requires at least 16 m$^3$ of He as a carrier gas for 1 yr of continuous $O_2/N_2$ observation, was unsuitable for use onboard NC2. Therefore,

we developed a low-flow system to perform in situ atmospheric $O_2$ and $CO_2$ observation onboard NC2. In this paper, we present details of the measurement system and report its fundamental performance in laboratory testing. We also discuss a problem that occurred when the measurement system was installed onboard NC2. Finally, we present 1 yr of atmospheric $O_2/N_2$, $CO_2$, and APO data and discuss the longitudinal distribution of the seasonal APO cycle in the North Pacific.

**2 Methods**

**2.1 Analytical system**

Figure 1 is a schematic diagram of the in situ observation system used onboard NC2. We used a fuel cell analyzer (Oxzilla-II, Sable System, USA) and a non-dispersive infrared analyzer (LI-840A, LI-COR, USA) for the onboard measurement of $O_2$ and $CO_2$, respectively. The sample air is drawn by a diaphragm pump (MOA-P108-HB, Gast

Mfg. Corp., USA) at a flow rate of about $8 \times 10^3$ cm$^3$ min$^{-1}$ and introduced into a spherical glass vessel with a volume of about $2 \times 10^3$ cm$^3$. Finally, the air is vented to the atmosphere through a back-pressure regulator (6800AL, Kofloc, Japan), which keeps the pressure inside the spherical vessel at about 0.05 MPa above ambient pressure. Water that condenses within the spherical vessel is drained from the bottom by a peristaltic pump (7016-21, Masterflex, USA). The sample gas for the $O_2$ and $CO_2$ measurements is drawn from the center of the spherical

vessel through 1/16-inch stainless steel (SUS) tubing. This mechanism based on a spherical glass vessel was adopted to reduce fractionation of the $O_2/N_2$ ratio (Yamagishi et al., 2008).

The sample gas from the spherical vessel is introduced into a two-stage cold trap (−80°C) to reduce the water vapor concentration to less than 1 ppm. Details of the cold trap are presented in Section 2.2. The dried sample gas and working reference gas, which is supplied from a high-pressure cylinder, are introduced into the two fuel cells of

the $O_2$ analyzer via two mass flow controllers (SEC-E40MK3, HORIBA STEC, Japan) and a 4-way 2-position valve (AC4UWE, Valco Instruments Co. Inc., USA). The mass flow controllers regulate the flow rates of the two gas streams with a precision of 0.01 cm$^3$ min$^{-1}$. The dried sample air and working reference air alternately pass through each fuel cell at intervals of 2 min by switching the 2-position valve. The $CO_2$ analyzer is placed





downstream of one of the fuel cells. The flow rates of the outflows from the $CO_2$ analyzer and the other fuel cell are monitored by mass flow meters (SEF-E40, HORIBA STEC, Japan) with a precision of 0.01 $cm^3$ $min^{-1}$. We adjusted the settings of the mass flow controllers to match the readings of the flow meters for the two air streams. Then the outflows of the flow meters are combined and vented to the atmosphere via a piezo actuator valve (PV-

1202MC, HORIBA STEC, Japan). The pressure downstream of the analyzers is actively controlled with respect to a reference pressure of a thermally insulated volume using the piezo actuator valve and a differential pressure sensor (Model 204, Setra Systems, USA).

Before the dried sample gas is introduced into the mass flow controller, it passes through a multi-position valve (EMTCSD6MWM, Valco Instruments Co. Inc., USA), to which three standard gases are connected. During the

calibration procedures, the multi-position valve selects the standard gases instead of the sample air, which is vented to the ambient air via a mechanical mass flow controller at a flow rate about 10 $cm^3$ $min^{-1}$. The 48-L aluminum cylinder for the working reference gas and the three 9.8-L aluminum cylinders for the reference gases are stored horizontally in thermally insulated boxes.

Custom software developed in LabVIEW (National Instruments Co., USA) and running on a PC controls valve

operation and the acquisition of digital data from the $O_2$ and $CO_2$ analyzers and analog data from the mass flow and pressure sensors.

### 2.2 Cold trap

Among the constituents of tropospheric air, water vapor content shows the widest range of variation, which causes apparent variations in the $O_2$ mole fraction of the air because the atmospheric $O_2$ is a major constituent of air (~21%).

For example, a water vapor increase of 1 ppm causes a 0.2-ppm apparent decrease in the $O_2$ mole fraction. Therefore, a two-stage cold trap was adopted to reduce the water vapor in the sample air to less than 1 ppm. Figure 2a shows the first version of the cold trap, which consisted of a free-piston Stirling cooler (FPSC) module (SC-UE15R, Twinbird, Japan), two disk-shaped aluminum blocks with grooves for 1/8-inch SUS tubes, and a cylindrical glass vessel with a volume of about $1.0 \times 10^3$ $cm^3$. The aluminum disks were placed between the cold head of the FPSC

module and the glass vessel and contact was tight. A platinum resistance thermometer was inserted into the aluminum block, and a temperature controller (E5CC, OMRON, Japan) regulated the FPSC module to maintain the aluminum blocks at −80°C. The sample air was dried by passing through the glass vessel first and then the SUS tube.





When the first cold trap was used for preliminary measurements at NIES during the summer (with high humidity), it worked without clogging for at least 1 month, which is the typical duration for a round trip using the North American route. However, the measurements were often interrupted because the SUS tube and inlet of the glass vessel clogged when it was used onboard NC2, and the frequency of clogging increased as the season progressed
from winter to spring to summer.

Thus, we changed the cold trap to the second design shown in Fig. 2b. In the second version of the cold trap, a cylindrical aluminum block with several holes is in contact with the cold head of the FPSC module, and a glass cold finger (volume of about $1.5 \times 10^3$ cm$^3$) and 1/8-inch SUS tube are inserted in the holes. The cold head and the aluminum block are insulated by a polyethylene resin cover. We began to use this cold trap for the shipboard
measurements in September 2016, and the cold trap has not clogged since that time.

**2.3 Calculation of $\Delta O_2$, $\Delta CO_2$, and $\delta(O_2/N_2)$**

The $O_2$ analyzer, equipped with two fuel cell sensors, was designed to precisely measure the difference in the $O_2$ mixing ratio between two air streams. Therefore, the change in the $O_2$ mixing ratio of the sample gas is determined as the relative change with respect to the working reference gas, which is supplied from the high-pressure aluminum
cylinder (48 L). In accordance with previous studies (e.g., Stephens et al., 2007; Thompson et al., 2007; van der Laan-Luijkx et al., 2010; Goto et al., 2013), the sample and reference air are alternately introduced into each fuel cell sensor by switching the 2-position valve at 1–5 min intervals. Figure 3a shows the temporal variation in the differential output signal of the $O_2$ analyzer during the measurement of reference gas from a high-pressure cylinder as a sample gas. Although the flow rate in this system (10 cm$^3$ min$^{-1}$) is lower than one-fourth of the flow rates
used in equipment in previous studies, the output signal shows an almost rectangular shape. The signal plateaus at least 1 min after the valve switching, and the output signal is averaged from the second minute of the cycle. The deviation of the $O_2$ mixing ratio in the sample gas from that of the working reference gas for the $i$-th 2-min interval, $\Delta O_{2,i}$, is computed based on the 1-min average according to the following equation:

$$\Delta O_{2,i} = (-1)^{(i-1)} \left[ v_i - (v_{i-1} + v_{i+1})/2 \right]/2 \tag{2}$$

where $v_i$ represents the average of the output signal for the $i$-th 2-min interval and the output signal represents the difference of the sample gas minus working reference gas when $i$ is an odd number.

In contrast to the $O_2$ analyzer, the $CO_2$ analyzer alternately measures the sample and working reference gases. The temporal variation of the output signal of the $CO_2$ analyzer for the same experiment shown in Fig. 3a is depicted in



Fig. 3b. As shown in the figure, the output signal does not plateau until after more than 90 s because of the relatively low flow rate in comparison with the volume of the optical cell of the LI-840A analyzer (14.5 cm$^3$). Therefore, we compute an average of the output signal for the last 10 s of each 2-min interval. Then, the deviation of the $CO_2$ mixing ratio of the sample gas from the working reference gas, $\Delta CO_{2,i}$, is computed according to

$$\Delta CO_{2,i} = (-1)^{(i-1)}[w_i - (w_{i-1} + w_{i+1})/2] \tag{3}$$

where $w_i$ represents the average for the $i$-th 2-min interval. Again, the sample gas is introduced into the $CO_2$ analyzer when $i$ is an odd number.

The variation in the atmospheric $O_2$ concentration is expressed as the change in the $O_2/N_2$ ratio with respect to an arbitrary reference value, and the $\delta(O_2/N_2)$ ratio is defined according to Eq. (1). Based on the $\Delta O_2$ and $\Delta CO_2$ values, $\delta(O_2/N_2)$ is given by the following equation:

$$\delta(O_2/N_2) = \frac{\Delta O_2}{S_{O_2}(1 - S_{O_2})} + \frac{\Delta CO_2}{(1 - S_{O_2})} \tag{4}$$

where $S_{O2}$ represents the $O_2$ mixing ratio in dry air ($S_{O2} = 0.2094$, Tohjima et al., 2005). In this calculation, we assume that only $O_2$ and $CO_2$ show more than ppm-level variation among all constituents of dry air, except nitrogen.

The time series of $\delta(O_2/N_2)$ and $\Delta CO_2$ calculated by Eqs. (2), (3), and (4) for sample gas provided from a high-pressure cylinder against the working reference air are plotted in Fig. 4. The standard deviations for $\delta(O_2/N_2)$ and the $CO_2$ mixing ratio are 3.8 per meg and 0.1 ppm, respectively, which likely correspond to the upper limits of precision because the measurements were taken in an air-conditioned laboratory.

## 2.4 Preliminary measurements of atmospheric O₂ and CO₂

We conducted preliminary observations of the atmospheric $O_2$ and $CO_2$ mixing ratios at Tsukuba, Japan, during the period July 10–17, 2015, to examine the performance of the $O_2$ and $CO_2$ measurement system. Outside air was drawn by the diaphragm pump from an air intake placed on top of our laboratory building. Two reference gases with high (−270 per meg) and low (−579 per meg) $O_2/N_2$ ratios were repeatedly introduced into the $O_2$ analyzer for 32 min each at intervals of 25 h. We determined a calibration line for $\delta(O_2/N_2)$ values from the measurements of the two reference gases during the observation. As for the $CO_2$ mixing ratios, a calibration line was determined from measurements of three reference gases with 387, 406, and 434 ppm before the observation. The $\delta(O_2/N_2)$ value and $CO_2$ mixing ratio were reported in our own original scales: NIES $O_2/N_2$ scale (Tohjima et al., 2008) and the NIES 09 $CO_2$ scale (Machida et al., 2011).





The observed $\delta(O_2/N_2)$ showed a diurnal cycle with an increase in daytime and decrease at nighttime. This cycle was inversely correlated with the $CO_2$ mixing ratio. The scatter plot between the $CO_2$ and $\delta(O_2/N_2)$ shows a clear negative correlation with the $\Delta O_2/\Delta CO_2$ slope of $-1.189\pm0.004$. During the observation, 32-min measurements of a check gas (CPD-00012: $-426$ per meg for $\delta(O_2/N_2)$ and 407.12 ppm for $CO_2$) supplied from an aluminum 10-L

cylinder were repeated twice daily. The $\delta(O_2/N_2)$ values and the $CO_2$ mixing ratios of the check gas showed steady values (Fig. 5); the average and the standard deviation ($\pm1\sigma$) were $-427.5\pm4.1$ per meg for $\delta(O_2/N_2)$ and $407.11\pm0.11$ ppm for $CO_2$. Moreover, there was no significant drift in the LI-840A analyzer during this observation. These results indicate the stability of the $O_2$ and $CO_2$ measurement system.

## 2.5 In situ measurements onboard NC2

In December 2015, the measurement system was installed in a deckhouse onboard NC2. An air intake was placed on a left-side deck rail of the navigation bridge, and air was drawn via a DK tube (NITTA, Japan) with an outer diameter of 10 mm and length of about 50 m. A 48-L aluminum cylinder for the working reference gas and three 10-L aluminum cylinders for the $O_2$ and $CO_2$ reference gases were placed in thermally insulated boxes, which were laid horizontally on a shelf to minimize the inhomogeneous distribution of $\delta(O_2/N_2)$ within the cylinders associated

with temperature and pressure gradients (Keeling et al., 1998, 2007).

The two reference gases with $-579$ per meg (CPD-00010) and $-270$ per meg (CPD-00011) were used for calibration of the $O_2$ analyzer. Since the $CO_2$ mixing ratios of these two reference gases are almost same ($\sim$407 ppm), we additionally used a third reference gas with a $CO_2$ mixing ratio of 448.3 ppm (CPB-17350) to calibrate the $CO_2$ analyzer. During the onboard measurements, these three reference gases were repeatedly measured for 32 min

every 24 h. To determine the calibration lines for both the $O_2$ and $CO_2$ analyzers precisely, the measurements of the three reference gases were repeated over 24 h when NC2 berthed at the port of Tahara, Japan.

## 3 Results and discussion

### 3.1 Influence of ship motion

After beginning the in situ measurements onboard NC2, we found that the ship motions did not affect the response

of the $CO_2$ analyzer but did seriously affect the response of the $O_2$ analyzer. Figures 6a and 6b show temporal variations in the output signal of the $O_2$ analyzer for the reference gas when NC2 was berthed at the port of Tahara and was cruising on the Pacific Ocean, respectively. The output signal during the cruise (Fig. 6b) shows apparent variations with peak-to-peak amplitudes of more than several tens of ppm and peak-to-peak periods of about 20 s.



Pickers (2016) reported that similar apparent variations caused by ship motion were superimposed on output signals of individual fuel cells equipped with an Oxzilla-II analyzer. However, in that case, the differential signal of both fuel cells did not show apparent variations because the motion-induced variations were almost compensated completely by the differential signals.

5 We installed a 3-dimensional accelerometer on the $O_2$ analyzer on March 3, 2016, to examine the relationship between the Oxzilla output signals and the ship's motion. The apparent variations in the output signal were associated with the variations in the acceleration of one axis or another and both amplitudes were roughly proportional to each other. However, we have not succeeded in simulating the apparent variations with a linear function of the measured accelerations. This is because the sensitivity of the $O_2$ analyzer to the acceleration along 10 the three axes seems to be unstable with time. Therefore, at this stage we cannot remove the apparent variations associated with the ship motions by a simple algorithm.

Figure 7 shows temporal variations in the $\delta(O_2/N_2)$ value of the two standard gases relative to the working gas during the 1-yr period of this study. In the figure, each blue circle represents the 32-min average of the standard gas during the voyages. The standard deviations of the 32-min averages were less than 13 per meg, suggesting that 15 the averaging procedure for several tens of minutes can effectively suppress the errors caused by ship motion. For example, the expected standard deviation of hourly averages of the $\delta(O_2/N_2)$ value is 9 per meg ($=13/2^{1/2}$).

The uncertainties of the 32-min averages of the standard gases are too large for calibration of the $O_2$ analyzer. In Fig. 7, the average $\delta(O_2/N_2)$ values of the standard gases determined when NC2 was berthed at the port of Tahara are also plotted as black circles. The error bars represent standard deviation. These plots are less variable than the 20 measurements during the voyages. Therefore, we calibrated the $O_2$ analyzer using interpolated calibration lines based on the results at the port just before each round-trip voyage.

### 3.2 Comparison between flask sampling and in situ observations

During the 1-yr period from December 2015 through November 2016, the in situ measurements of the atmospheric $O_2$ and $CO_2$ concentrations were conducted during nine round-trip voyages, from NC2-123 to NC2-131, along the 25 North American route. The individual cruise tracks are depicted in Fig. 8a, where thin lines correspond to intervals of missing measurements. We obtained no in situ data during the two westbound voyages of NC2-123 and NC2-128 because the cold trap became clogged. Along with the in situ measurements, air samples were collected in seven 2.5-L glass flasks at fixed longitudes (130°W, 145°W, 160°W, 175°W, 170°E, 155°E, and 145°E) during each westbound cruise.



The time series of $CO_2$, $\delta(O_2/N_2)$, and APO data taken from the in situ measurements and the flask samplings are shown in Fig. 9. The APO is computed based on the $\delta(O_2/N_2)$ value and $CO_2$ mixing ratio in accordance with the following equation:

$$\delta APO = \delta(O_2/N_2) + 1.1 \times \frac{X_{CO_2}}{S_{O_2}} - 1850 \qquad (5)$$

where 1850 is an arbitrary APO reference point adopted by NIES. In Fig. 9, each point for the in situ measurement represents the hourly average and the data outside the longitudinal range between 140°E and 128°W are excluded because those data are significantly contaminated by anthropogenic emissions from the coastal regions of both Japan and North America. The time series in Fig. 9 clearly shows seasonal cycles and the in situ data seem to agree with the flask data. The differences in the $CO_2$, $\delta(O_2/N_2)$, and APO values between the in situ and the 39 flask measurements (in situ – flask) are depicted in Fig. 10. The averaged differences with standard deviations were −0.02±0.33 ppm of $CO_2$, −2.1±9.2 per meg of $\delta(O_2/N_2)$, and −2.2±9.3 per meg of APO. Taking into account the uncertainties of the flask measurements (0.05 ppm for $CO_2$ and 5 per meg for $O_2/N_2$ measurements, Tohjima et al., 2003), we conclude that these standard deviations correspond to the upper limits of the uncertainties of the in situ measurements onboard NC2. The differences between flask sampling and in situ measurements by the GC/TCD method were reported as −0.6±9.1 per meg of APO on the Oceanian route (Tohjima et al., 2015), and 7.0±9.9 per meg of $\delta(O_2/N_2)$ at Cape Ochi-ishi (Yamagishi et al., 2008). From these results, we conclude that the reliability of the $O_2$ measurements of this study is similar to that of the GC/TCD method.

The time series of the in situ data shown in Fig. 9 do not necessarily show smooth changes, which may be partly attributed to the fact that the onboard observations were conducted in the broad area of the North Pacific. For example, the APO values show relatively noticeable decreases during the eastbound voyage NC2-125 in early March and increases during the eastbound voyage NC2-127 in late May. The longitudinal distributions of 5-h running average of APO for the individual round-trip cruises are depicted in Fig. 8b, which clearly shows these anomalously low and high APO distributions. From the preliminary analyses, these anomalies seem to be explained by air mass transport and the expected air-sea gas exchanges in the source regions. Detailed discussions of the anomalous changes are beyond the scope of this paper and will be presented in future work.

### 3.3 Distribution of seasonal cycles in the North Pacific

We investigated the longitudinal distribution of the seasonal amplitude of APO in the middle latitudes of the North Pacific using the 1 yr of in situ data within the area of 29°N–45°N and 140°E–130°W (Fig. 8a). The in situ data





were binned into 10 longitudinal bands (140°E–150°E, 150°E–160°E, …, 140°W–130°W) and fitted to the following function using a least-squares method:

$$f(t) = a_0 + a_1 t + \sum_{i=1}^{2} [a_{2i} \sin(2\pi i t) + a_{2i+1} \cos(2\pi i t)] \tag{6}$$

where $a_1$ is a detrending coefficient ($-7.9$ per meg yr$^{-1}$) based on an average decreasing trend of APO values measured from the flask samples collected onboard NC2 during the 4-yr period 2014–2016.

The detrended in situ data for the individual longitudinal bins and the fitted average seasonal cycles are shown in Fig. A1. The APO seasonal amplitudes for the 10 bins are plotted along with the longitude in Fig. 11a. For comparison, we also plot seasonal amplitudes of APO in the North Pacific reported by Tohjima et al. (2012). In the previous study, APO data from the flask samples collected in the Pacific during the period from 2002 to 2008 were binned into several rectangular regions and the seasonal cycles for the binned data were examined in the same way as in this study. The seasonal amplitudes in the previous study varied from 20 per meg to 110 per meg, while those in this study varied from 51 per meg to 73 per meg. The difference in the seasonal amplitudes seems to be explained by the dependence of the latitudinal distribution, which is clearly shown in Fig. 11b, where all the seasonal amplitudes shown in Fig. 11a are plotted along with the latitude. Such latitudinal dependence of the APO amplitude was previously pointed out for data from the western Pacific (Tohjima et al., 2005, 2012). On the other hand, this study reveals that the longitudinal variability in the seasonal amplitude of APO in the North Pacific is rather small. This preliminary analysis suggests that the temporally and spatially dense atmospheric $O_2$ and $CO_2$ data obtained from the in situ observation onboard NC2 will enhance our understanding of air-sea gas exchange in the North Pacific.

## 4 Conclusion

We developed a ship-borne system to continuously measure atmospheric $O_2$ and $CO_2$ concentrations based on a fuel cell oxygen analyzer (Oxzilla-II) and a non-dispersive infrared $CO_2$ analyzer (LI-840A). To reduce the consumption rate of working reference gas and standard gases supplied from high-pressure cylinders, a relatively low flow rate of 10 cm$^3$ min$^{-1}$ was adopted for the measurement system. By keeping the flow rate and pressure of the system constant, we achieved precisions of 3.8 per meg for the $O_2/N_2$ ratio and 0.1 ppm for the $CO_2$ measurements in the laboratory.



We installed the measurement system on commercial cargo ship NC2 and started onboard continuous measurements in December 2015. We found that the ship motion significantly affected the output signal of the $O_2$ analyzer; apparent wavy variations with amplitudes of more than 20 ppm and peak-to-peak periods of about 20 s were superimposed on the output signals during the voyage. Although this variation has not been eliminated yet, 1-h averaging considerably suppresses the -variation associated with the ship motion because of the oscillatory nature of the apparent variations. From the comparison between the in situ measurements and simultaneously collected flask samples, we concluded that the uncertainties of $\delta(O_2/N_2)$ and $CO_2$ concentration for the in situ measurements are about 9 per meg and about 0.3 ppm, respectively.

Using the in situ data obtained during the 1-yr period from December 2015 to November 2016, we examined longitudinal (140°E–130°W) distribution of seasonal APO amplitude at the middle latitudes (29°N–45°N) in the North Pacific. The amplitudes showed rather small longitudinal variability, with a range from 51 per meg to 73 per meg in comparison to the latitudinal variations reported by Tohjima et al. (2012). Although the problem related to motion-induced degradation of $O_2$ measurement precision has not been resolved, this study clearly demonstrated that in situ observation onboard cargo ships can extend the coverage of the atmospheric $O_2$ and $CO_2$ data to a degree that flask sampling could never achieve.

**Acknowledgment**

We gratefully acknowledge the generous cooperation of Toyofuji Shipping Co. and Kagoshima Senpaku Co. for providing us the opportunity to make the onboard atmospheric observations. Thanks are also expressed to the crew of *New Century 2*. We would like to thank Tomoyasu Yamada, Nobukazu Oda, and other members of the Global Environmental Forum for their continued support in maintaining the $CO_2$ and $O_2$ measurement system. We thank Hisayo Sandanbata, Eri Matsuura, and Motoki Sasakawa of the NIES for their continued support in the $O_2/N_2$ and $CO_2$ analysis of flask samples. This work was financially supported by a Grand-in-Aid for Scientific Research, and in part by the Global Environmental Research Coordination System, from the Ministry of the Environment, Japan (E1451).



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





**Figure 1: Schematic diagram of atmospheric O₂ and CO₂ measurement system used onboard cargo ship NC2.**

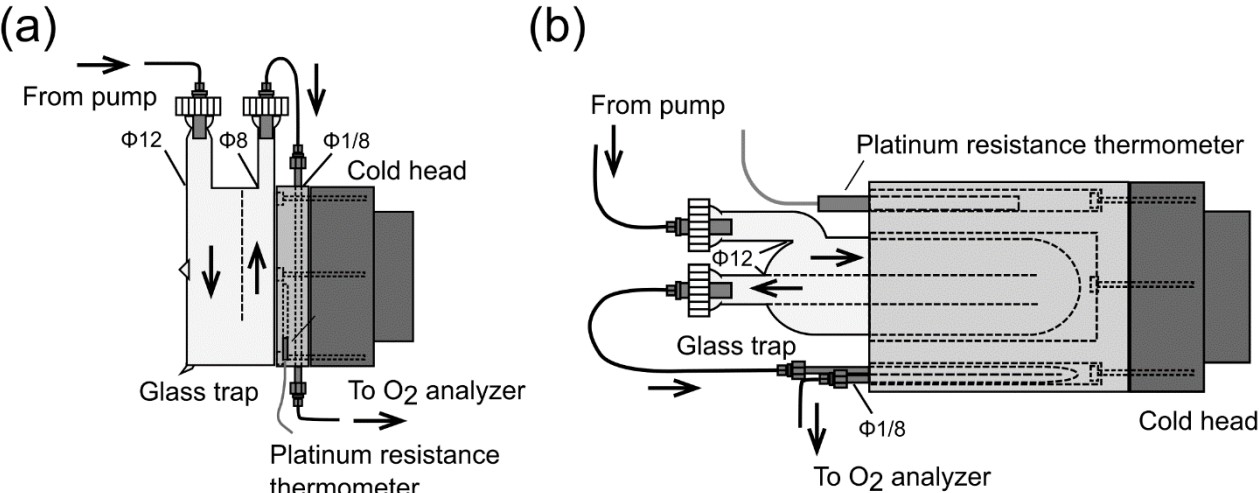

**FIgure 2: Schematic diagram of (a) first and (b) second versions of cold trap for reducing water vapor in samples.**



**Figure 3: Temporal variations in the output signals of (a) O₂ analyzer and (b) CO₂ analyzer when reference air from a high-pressure cylinder was measured as sample air.**



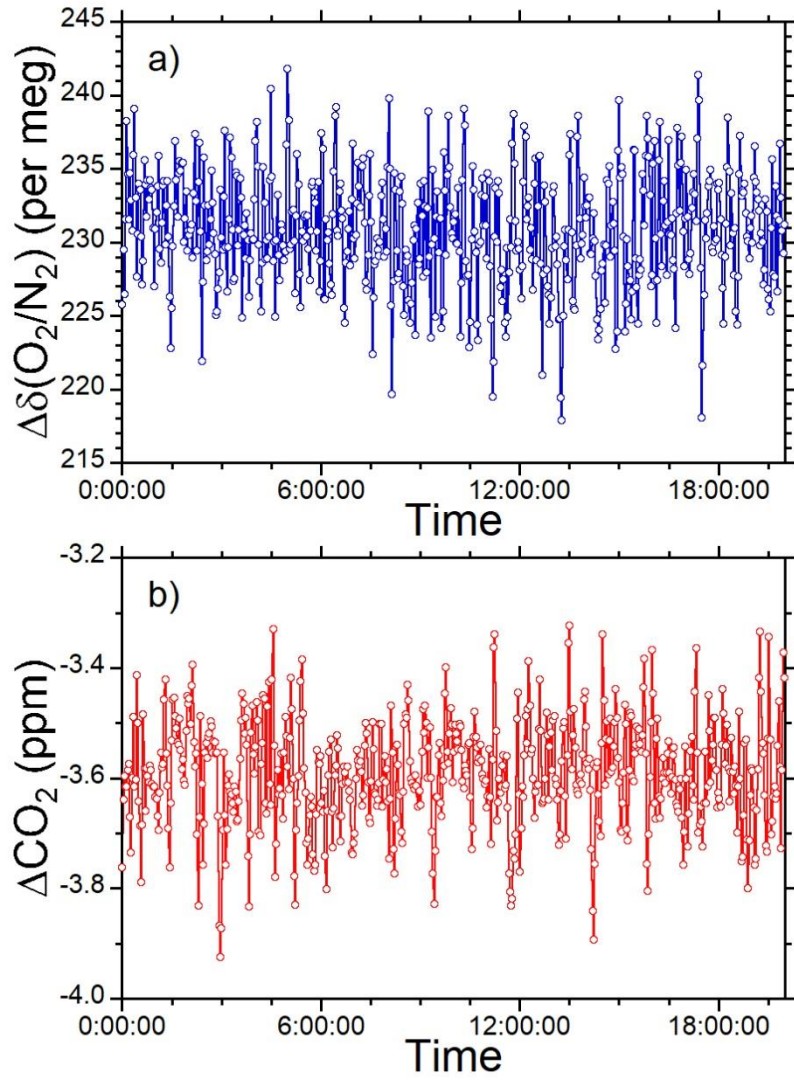

**Figure 4: Time series of δ(O₂/N₂) and ΔCO₂ calculated by Eqs. (2), (3), and (4) for sample air provided from high-pressure cylinder against the working reference.**



**Figure 5: Time series of δ(O₂/N₂) (blue, left axis) and CO₂ mixing ratio (red, right axis) observed at Tsukuba during July 10–17, 2015. The δ(O₂/N₂) and CO₂ mixing ratio of the periodically measured check gas are also depicted as light blue and pink circles, respectively.**



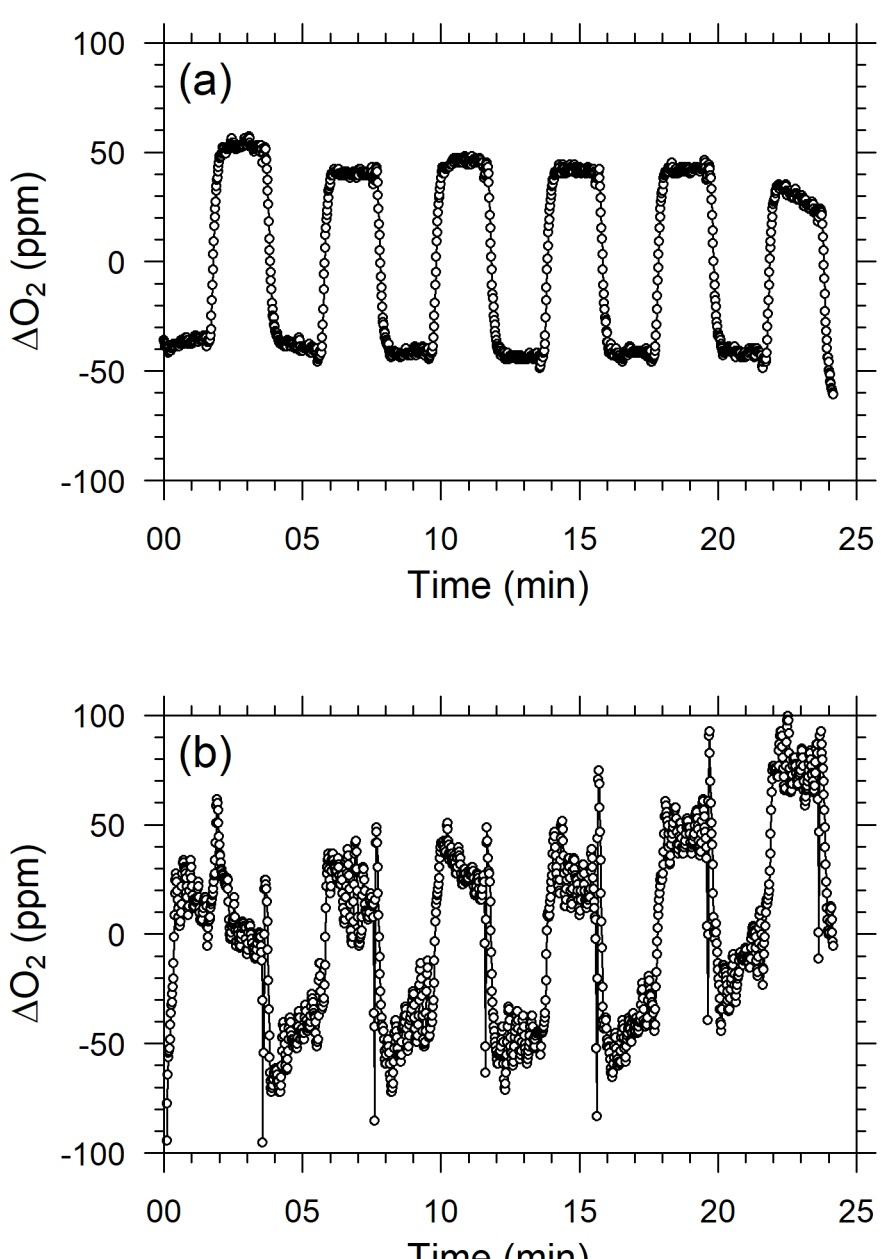

**Figure 6: Temporal variations in differential output signals of O₂ analyzer for six measurement cycles of O₂ reference gas while ship was (a) in harbor and (b) cruising.**



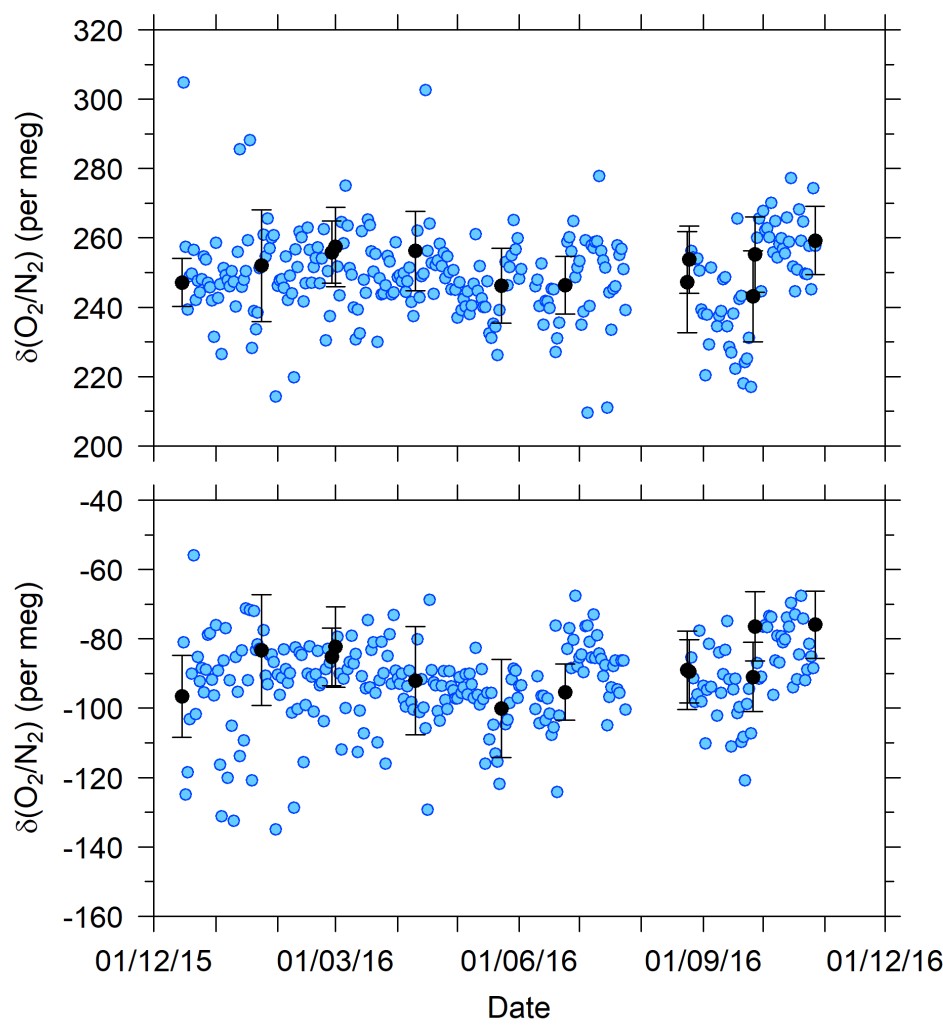

**Figure 7: Time series of differences in δ(O₂/N₂) for two reference gases (top) CPD-00011 and (bottom) CPD-00010 with respect to working reference gas, as determined onboard NC2 during the 1-yr period of this study. Blue circles represent 32-min average values of reference gas measurements carried out at 24-h intervals. Black circles represent the averages of calibrations conducted when NC2 was berthed at the port of Tahara, and the error bars represent standard deviations.**



**Figure 8: (a) Cruise tracks of NC2 for nine round trips (NC2-123, NC2-124, …, NC2-131) during the period from December 2015 to November 2016. Thin lines represent intervals where in situ measurements onboard NC2 were interrupted. (b) Longitudinal distribution of hourly APO taken from the nine cruises. Five-hour running averages are applied to the hourly data to reduce signal noise.**





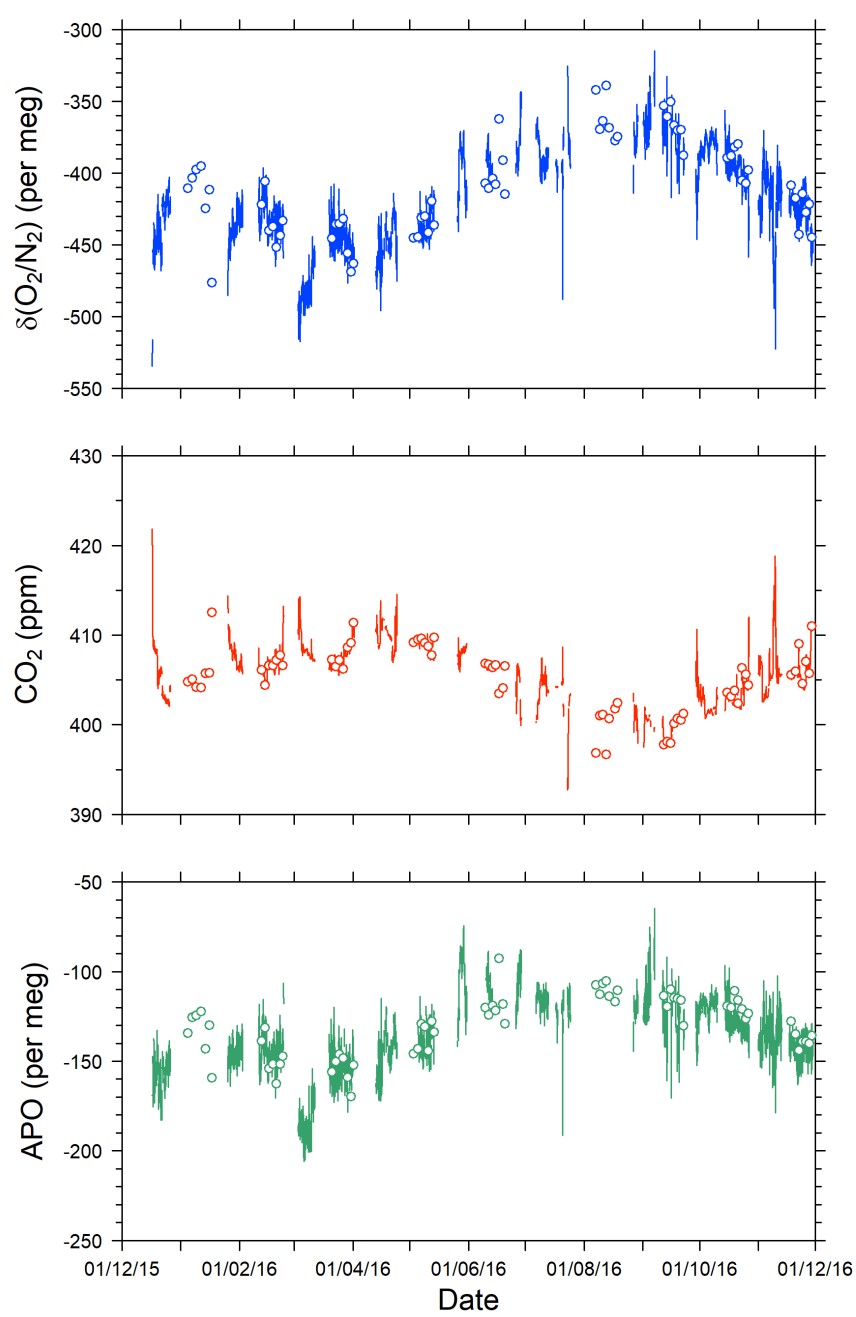

**Figure 9: Time series of (top) δ(O₂/N₂), (middle) CO₂ concentrations, and (bottom) APO during 1-yr period from December 2015 to November 2016. Lines indicate continuous observation, and circles indicate flask measurements. Data obtained near the coasts of Japan and North America are excluded.**



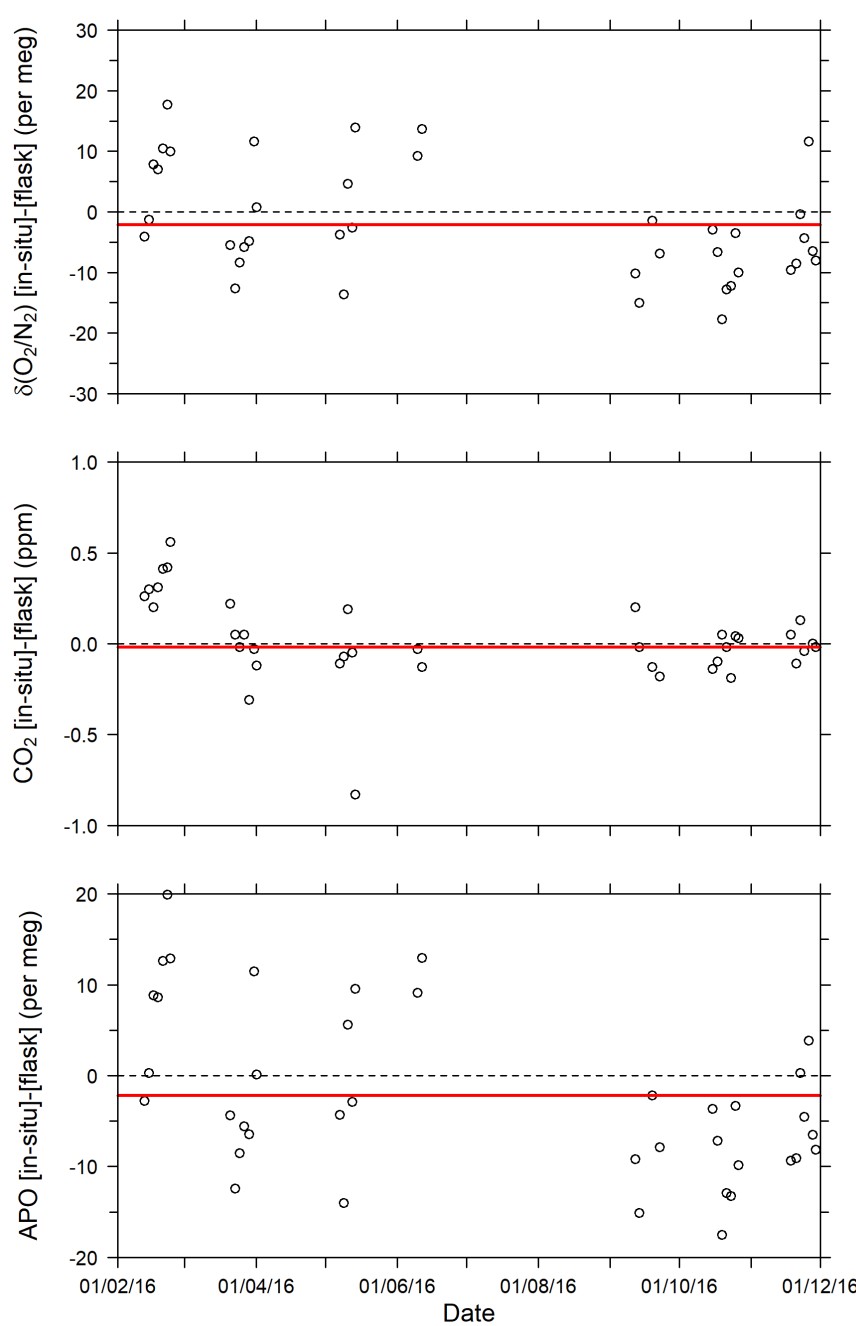

**Figure 10:** **Time series of differences of (top) δ(O₂/N₂), (middle) CO₂, and (bottom) APO between the in situ measurements and flask measurements (in situ – flask). The red lines show average values.**



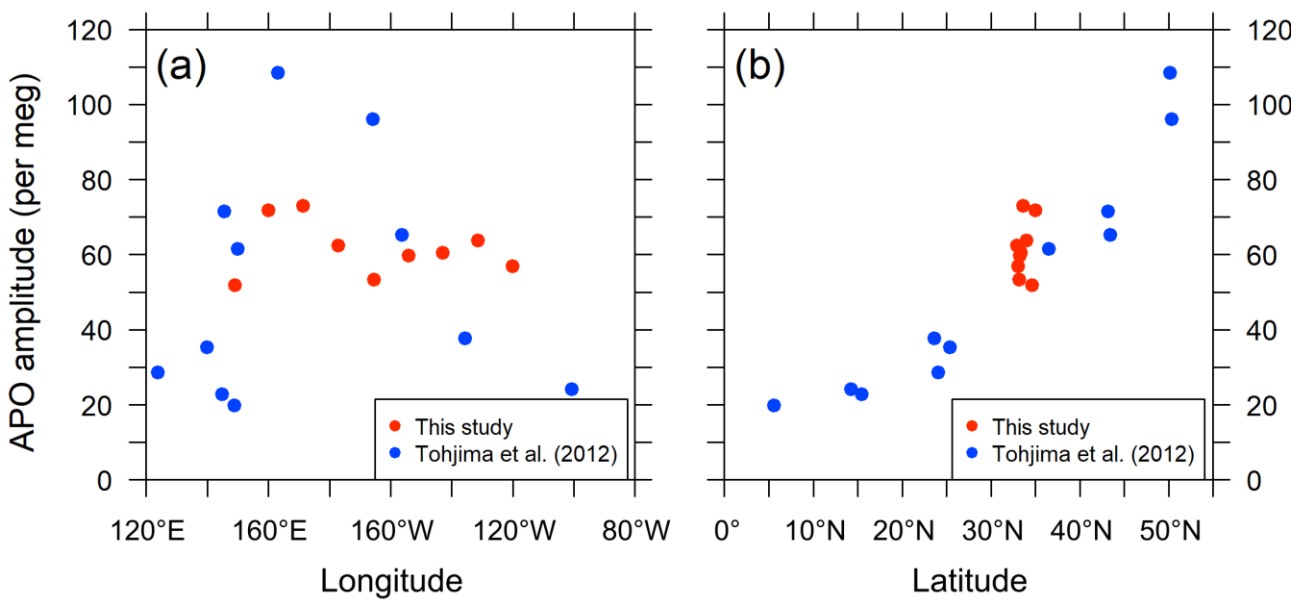

**Figure 11: Distribution of seasonal amplitudes of APO along with (left) longitude and (right) latitude in the North Pacific. The red symbols represent APO amplitudes of the in situ data of this study for individual longitudinal bins within middle latitudes (29°N–45°N and 140°E–130°W). The blue symbols represent APO amplitudes reported by Tohjima et al. (2012). The longitudes and latitudes of the plots are the average positions of the data within the individual bins.**



## Appendix: The averaged seasonal cycles of APO



**Figure A12: Detrended seasonal variations of APO for the 10 longitudinal bins within the rectangular area of 29°N–45°N and 140°E–130°W. Red circles represent hourly averages of in situ APO data. Solid lines are average seasonal cycles determined by a least square method. Dashed lines indicate 95% confidence intervals determined by linear prediction model.**