# Peer review of "In situ* observation of atmospheric oxygen and carbon dioxide in the North Pacific using a cargo ship"

_Atmospheric Chemistry and Physics, 2018_

## Referee Comment (RC1) · Anonymous Referee #1 · 30 Mar 2018

[General comments]

In this paper, the authors present new continuous observations of atmospheric $O_2$ and $CO_2$ in the North Pacific using a cargo ship for the period December 2015 – November 2016. Since continuous $O_2$ measurements are still limited globally, the results and know-how presented in the paper would give a valuable contribution to the understanding of carbon cycle and air-sea gas exchange. The manuscript is well written and can be accepted with only minor revisions.

[Specific comments]

1) P3, L3–5: Authors should clarify the reason why "a change of $O_2$ per mol of dry

air does not necessarily result in a 1-ppm change in the O2 mole fraction but always corresponds to a 4.77 per meg change in the $\delta$(O2/N2) value". And/or please add the appropriate reference(s).

2) P4, L14 "The sample air is drawn by a diaphragm pump...": It is better to add the information of filter. What kind of filter did you use? (material, mesh size... etc.)

3) P5, L9 "three standard gases": Are these "standard gases" same as "reference gases" on page5, line 12?

4) P6, L17 "1–5 min intervals": According to page 4, line 4, I understood that the switching interval is 2 min. What do the "1–5 min intervals" mean? Did you test the switching intervals from 1 min to 5 min and decide it 2 min?

5) P7, L15–16: How many hours of data did you use for the calculation of the standard deviations? 1-h? 24-h? Please clarify it in the text.

6) P8, L3: It would be better to mention what the slope value of -1.189$\pm$0.004 means.

7) P8, L4 and L13 "10-L cylinder": Are these 10-L cylinders different from "9.8-L cylinder" on page 5, line 12?

8) P10, L10–11: Please clarify the time period for averaging. It seems that the differences from February to June in each figure are scattered around zero, but the differences in $\delta$(O2/N2) and APO from September to November look shifting downward. Are there any possibilities that the differences between the in-situ data and flask data are temporally changing? Is it negligible because of uncertainty?

9) Some expressions of O2 are used in the manuscript, but I couldn't catch the difference. For example, authors use "O2/N2 ratio" on page 3 (line 12), but "These O2 and ...", "...continuous O2/N2 observation...", and "...the $\delta$(O2/N2) ratio is..." are used on page 3 (line 15), page 4 (line 4), and page 7 (line 9), respectively. These expressions should be reconsidered throughout the manuscript. Similarly, the expressions of CO2 should also be reconsidered throughout the manuscript. For example, "CO2 mixing

ratio" (e.g. page 8, line 18) and "CO2 concentration" (e.g. page 9, line 24) are used in the manuscript.

Technical corrections:

1) P2, L18: Change ":" after Naegler et al., 2007 to ";".

2) P4–5, 2.1 Analytical system: Uniform the names of parts in the system in the text and Figure 1. For example, "glass vessel", "4-way 2-position valve", and "piezo actuator valve" are used in the text, but these are showed as "glass flask", "2-position valve", and "variable valve" in Figure 1.

3) P8, L1: I think it would be better to add some words to make the readers focus to Figure 5. For example, "As shown in Fig. 5, ".

4) P10, L1 and 9: I think it would be better to switch the order of CO2 and $\delta$(O2/N2).

5) P12, L5: Remove "- (hyphen)" from "the -variation".

6) Units in section 2: Units of "cm3 min-1" and "cm3" are used as flow rate and volume in the text, but those in Figure 1 are "mL/min (or L/min)" and "L". Please uniform the units throughout the manuscript.

7) Figure 4 a: I think "$\Delta$" in the label of vertical axis should be removed.

8) Figure 6 b: It is not clear the apparent variations of several tens of ppm amplitudes and 20s intervals in this figure. It would be better to add the expanded figure of apparent variations.

9) Figure 9: It would be very informative to add the cruise information in this figure. For example, changing the color depending on cruises, adding cruise-name labels. . .etc.

10) Figure A1: Modify from "Figure A12" to "Figure A1".

---

## Referee Comment (RC2) · Anonymous Referee #2 · 23 Apr 2018

In this manuscript, Hoshina et al. describe the construction, installation and performance of a fuel-cell/NDIR-based instrument for measuring O2 and CO2 aboard ships of opportunity. They also present a year of data collected while underway across the North Pacific. Overall, this is a very straightforward presentation of careful work. Nothing about this project is revolutionary or ground-breaking, but the continued development of O2-CO2 systems is important for the advancement of atmospheric potential oxygen (APO) as a useful oceanographic tool, and the data they have collected fill an significant gap in the APO community's records. This is valuable work and definitely deserves to be published. I have very few scientific questions or concerns about the content. With many years of excellent work in this field, the Tohjima lab knows what it

is doing, and this paper reflects that expertise. It is also quite well written.

======================================================================

Questions/comments on substance

P3 line 3: Change to "The mole fraction is not used as a measure of O2 abundance because, for example, a change" Furthermore, it would be helpful if the authors gave an explanation of why mole fraction changes when per meg doesn't. This is alluded to on page 7 (around eq. 4), but it should be stated more explicitly at the outset.

P5 line 5-6: I believe you are trying to say that the outlet pressures (and by extension, the pressures in the analysis cells), are kept at the same absolute value at all times by actively matching them to a reference volume. If this is what you do in fact mean to say, you should probably make it a bit more explicit.

P5 line 11: Is the "mechanical mass flow controller" the same as the one mentioned on P4 line 25? If so please make it clearer. If not, specify this one more completely.

P5 line 23: Is there more than one SUS tube? If so, explain how many and why.

P6 line 10: To what do you attribute the improved performance of the 2nd trap design? Is it simply larger glass volume, more complete chilling of the glass due to the insulation, the coaxial design of the glass trap...? Please give a few words of explanation.

P6 line 17: At what interval did you operate your changeover valve? "1-5 minutes" is not specific enough, as the frequency of these switches can have a significant impact on the precision of your results (see, for example, Keeling et al, 2004). How did you settle on this interval?

P6 line 20-21: It's not at all clear from the plot that the CO2 ever reaches a plateau before the changeover valve changes state again. My concern is that the CO2 values you infer will be consistently biased toward the value of whatever gas was previously in the analysis chamber. Perhaps a better approach is to fit each transition with function

like (1 – exponential), with 2 free parameters (rise time and asymptote) and use the asymptote as the true mixing ratio, even this value is never measured in the instrument.

P6 lines 26: What happens when i=1? I recognize that this formula represents the difference between a block and the average of the blocks before and after, but the formula doesn't work for the first block after a calibration run.

P7 line 24-25: I'm a bit confused by your calibration procedure. I think you're saying that you used two span tanks for O2 and calculated a linear response function for the Oxzilla every 25 hours. How did you interpolate the response of the instrument for the many observations that were made during the intervening 23.5 hours? Similarly, it seems like you had a 3-point calibration for CO2 made just once at the beginning of the week of observations. If you have a 3-point calibration, did you assume a linear response function, or did you allow a 3-parameter form with some curvature? Am I correct that you did not repeat the CO2 calibration at any time during or after the 7-day period of observations? Please clarify these points.

P8 line 19: I think you mean "During every 24hr period, these three reference gases were measured for 32min each." If that's right, please reword. If not, clarify.

P8 line 28: There is no way for us to tell (from Fig. 6b) that there are 20s cycles. It would be very helpful if there were an inset figure that zoomed in on a subset of the data with a much-expanded time-scale.

P9 line 15-16: Are the "hourly averages" mentioned on line 16 tank values, or atmospheric data? If the latter, wouldn't natural variability in the atmosphere lead you to expect more scatter? See also my comments on Fig. 7 below.

P10 lines 10-14: I am puzzled about why the authors didn't just deconvolve the flask errors to get a separate value for the uncertainty in the shipboard data. If the (shipboard data – flask values) have a scatter of XX per meg and the flasks themselves have an uncertainty of YY, then isn't the uncertainty in shipboard data alone just given by ZZ =

sqrt(XXˆ2 – YYˆ2)? This way one can report a meaningful error for the system, rather than an upper limit.

P10 line 20: I cannot see the "noticeable decrease" in Figure 9. Perhaps a zoomed-in inset would show it. In fact, when I look at Figure 8, I think I seen an _increase_ in APO during the eastbound leg of NC2-125.

P11 line 5: The period 2014-2016 is not 4 years in length. Please correct either the dates or the duration.

P11 line 7: Presumably the figure referenced here is the one labelled "A12". It's odd to me that this one figure would comprise an entire appendix. I would suggest simply making it a part of the normal figures (since it is a valuable one). At the very least, make sure it is correctly referenced in the text.

Figure 3: This would be much more valuable if there were fewer cycles shown with more detail. Perhaps an inset with one representative cycle (for each of plots "a" and "b") would make the true stability of the measurements more apparent.

Figure 7: I am puzzled that the error bars on the black circles (when the ship is in port) are not apparently any smaller than the scatter in the blue points (taken at sea). I'm just going by eye, but it certainly looks like the error bars in port capture more than 68% of the blue points. Wouldn't the rocking of the ship and the resulting response in the Oxzilla II make the scatter a bit larger when at sea? In lines 19-20 you say the scatter is actually bigger in the blue points, but I just can't see that. Also, on lines 20-21, you say you are only using the in-port measurements for calibration of the Oxzilla. Really? What if the instrument response varies during the time between ports? Why not use the information from the tank runs taken at sea (the blue points in Figure 7) to address this possibility.

===========================================================================

Strictly editorial comments:

Throughout the mansucript: Please italicize "in situ"

Throughout the manuscript: Please change "onboard" to "aboard"

P1 line 6: Change "observed" to "measured"

P1 line 19: Change to "compared the year of"

P2 line 11-12: Change to "the introduction of the tracer atmospheric"

P2 line 14: Remove "the" (the first word of the line)

P2 line 20-21: Change to "about 21%. Keeling (1988) was the first to develop an atmospheric O2 measurement technique with this precision, using an interferometer and showing the usefulness"

P3 line 2: Should read "Keeling and Shertz"

P3 line 17: Change to "trasects of the data in the western Pacific region revealed that variation in the magnitude of the bulge in annual mean APO"

P3 line 18-20: Change to "Tohjima et al., 2015). This analysis was made possible by the relatively high spatiotemporal sampling desnity in the western Pacific."

P3 line 21: Change to "route made it"

P3 line 25: Change to "Therefore, in December 2015 we initiated a program of in situ measurements aboard"

P3 line 26: Remove "was also desired"

P3 line 28: For consistency, this should read "along the North American route in the Pacific."

P4 line 1: Change to "since it is difficult to load and unload"

P4 line 2: Remove "are cumbersome"

P4 line 3: Change to "frequency. With these constraints the GC/TCD"

P4 line 5: Change to "and CO2 measurements aboard NC2."

P4 line 16: Change to "cm3. The air"

P4 line 20: Change to "tubing. The technique of sampling from a spherical"

P4 line 22: Change to "The gas sampled from the spherical vessel"

P5 lines 3-4: Change to "controllers until the readings of the flow meters for the two air streams matched. The outflows of the"

P5 line 18: Change to "water vapor shows"

P6 lines 12-13: Change to "gas is reported as a relative"

P6 line 18 : Change to "analyzer during a test run in which a reference gas from a high-pressure cylinder was used as a sample gas."

P6 line 19-20: Change to "min-1) is more than four times slower than the flow rates used in previous studies, the"

P6 line 25: Change to "signal for the second minute of the i-th 2-minute interval"

P7 line 6: Change to "the average of the last 10s of data for the i-th 2-minute interval."

P7 line 14: Change to "Eqs. (2), (3), and (4) for "sample" gas provided"

P7 lines 16-17: Change to "which likely represents the best possible precision, because the"

P8 line 1: Change to "O2/N2) shown in Figure 5, reveals a diurnal cycle"

P8 line 2: Change to "ratio. A scatter plot of CO2 and"

P8 line 16: Change to "-579 mer peg (tank #CPD-00010)"

P8 line 19: Change to "During the shipboard measurements"

P9 line 2: Change to "fuel cells in an Oxzilla-II analyzer. However, in her instrument, the differential"

P9 line 8: Change to "succeeded in describing the"

P9 line 11: Change to "motions with a"

P10 line 7: Change to "because of significant contamination by anthropogenic"

P10 line 21: Change to "distributions of the 5-hr running"

P10 lines 23-24: Change to "APO distributions. Preliminary analysis of the cause of these anomalies points to atmospheric transport and the"

P10 line 25: Change to "will be presented in a future publication."

P11 line 4: Change to "-7.9 per meg yr-1) determined from the APO values"

P12 line 5: Remove the unneeded "-" before variation

P12 line 22: Change to "Grant-in-Aid"

---

## Author Comment (AC1) · 8 Jun 2018

[General comments]

*In this paper, the authors present new continuous observations of atmospheric O2 and CO2 in the North Pacific using a cargo ship for the period December 2015 – November 2016. Since continuous O2 measurements are still limited globally, the results and know-how presented in the paper would give a valuable contribution to the understanding of carbon cycle and air-sea gas exchange. The manuscript is well written and can be accepted with only minor revisions.*

We would like to thank the anonymous referee for his/her careful reading of our paper and helpful comments. We have revised the manuscript and describe the changes in the following. The referee's comments are in *blue italics*, our responses to the specific comments below.

[Specific comments]

*1) P3, L3–5: Authors should clarify the reason why "a change of O2 per mol of dry air does not necessarily result in a 1-ppm change in the O2 mole fraction but always corresponds to a 4.77 per meg change in the (O2/N2) value". And/or please add the appropriate reference(s).*

To respond the comments from both reviewer #1 and #2, we have change the relevant sentence "The reason for not … in the $\delta(O_2/N_2)$ value" to "The mole fraction is not used as a measure of $O_2$ abundance because the changes in the mole fraction of major atmospheric constituents like $O_2$ are sometimes very confusing. For example, adding 1 µmol of $O_2$ to an air parcel containing 1 mol of dry air results in a 0.79-ppm increase in the $O_2$ mole fraction and adding 1 µmol of $CO_2$ results in not only a 1-ppm increase in the $CO_2$ mole fraction but also a 0.21-ppm decrease in the $O_2$ mole fraction. These confusing results are attributed to influences of the changes in the total number of moles in the air parcel on the mole fractions or dilution effect (e.g., Keeling et al., 1998; Tohjima 2000). However, adding 1 µmol of $O_2$ to 1 mol of dry air, which contains 0.2094 mol of $O_2$ (Tohjima et al., 2005), always results in a 4.77-per meg change in the $\delta(O_2/N_2)$ value." (P3 L 3-10).

*2) P4, L14 "The sample air is drawn by a diaphragm pump. . .": It is better to add the information of filter. What kind of filter did you use? (material, mesh size. . . etc.)*

We used a polypropylene cartridge filter with a mesh size 7 µm (MCP-7-C10S, ADVANTEC, Japan). The relevant sentence has been changed to "After passing a polypropylene cartridge filter with a mesh size of 7 µm (MCP-7-C10S, ADVANTEC, Japan), the sample air is drawn by a diaphragm pump…" (P4, L19-20).

*3) P5, L9 "three standard gases": Are these "standard gases" same as "reference gases" on page5, line 12?*

Yes, the "three standard gases" are same as "reference gases". To unify the term throughout the manuscript, we have changed "reference gases" to "standard gases".

*4) P6, L17 "1–5 min intervals": According to page 4, line 4, I understood that the switching interval is 2 min. What do the "1–5 min intervals" mean? Did you test the switching intervals from 1 min to 5 min and decide it 2 min?*

To respond the reviewers' comments, we have changed the ambiguous descriptions to "In previous studies (e.g., Stephens et al., 2007; Thompson et al., 2007; van der Laan-Luijkx et al., 2010; Goto et al., 2013), the sample and reference air are alternately introduced into each fuel cell sensor by switching the 4-way 2-position valve at 1- to 5-min intervals. In this study, we adopted 2 min for the valve-switching intervals in light of the responses of the $O_2$ and $CO_2$ analyzer after valve switching, as described below." (P6 L 22-25).

*5) P7, L15–16: How many hours of data did you use for the calculation of the standard deviations? 1-h? 24-h? Please clarify it in the text.*

We used 20 hours of data to calculate the standard deviations. We have added this information in the sentence as follows: "The standard deviations for $\delta(O_2/N_2)$ and $\Delta CO_2$ calculated from 20 h of data are 3.8 per meg and 0.1 ppm …". (P7 L24-L26).

*6) P8, L3: It would be better to mention what the slope value of -1.189±0.004 means.*

To respond to the reviewer's comment, we have changed the sentence "The scatter plot between … slope of −1.189±0.004" to "A scatter plot of $CO_2$ and $\delta(O_2/N_2)$ shows a clear negative correlation with the $\Delta O_2/\Delta CO_2$ slope of −1.189±0.004, which is close to the land biotic $O_2$ to $CO_2$ exchange ratio of -1.10±0.05. Since the observation was conducted in summer and coal consumption is limited in Tsukuba, the $\Delta O_2/\Delta CO_2$ slope means that the observed $CO_2$ changes can be predominantly attributed to the activity on land." (P8 line 12-16).

*7) P8, L4 and L13 "10-L cylinder": Are these 10-L cylinders different from "9.8-L cylinder" on page 5, line 12?*

These 10-L cylinders are same type as 9.8-L cylinder. To unify the term, we have changed "9.8-L cylinder" to "10-L cylinder" through the manuscript.

*8) P10, L10–11: Please clarify the time period for averaging. It seems that the differences from February to June in each figure are scattered around zero, but the differences in (O2/N2) and APO from September to November look shifting downward. Are there any possibilities that the differences between the in-situ data and flask data are temporally changing? Is it negligible because of uncertainty?*

Although checking the shipboard data and flask data carefully, we haven't determined the reason of the apparent downward shifts of the in-situ data from the flask data for the voyages NC2-129 and NC2-130. We think changes in the response functions of the oxygen analyzer would at least partly explain the shift of the differences, but it is difficult to determine the changes in our measurement conditions during voyages. However, taking the uncertainty (1σ) of the differences between in-situ and flask measurements, 9 per meg, we conclude that the differences for the voyage NC2-129 and NC2-130 are negligible because those data are within 2σ (18 per meg).

*9) Some expressions of O2 are used in the manuscript, but I couldn't catch the difference. For example, authors use "O2/N2 ratio" on page 3 (line 12), but "These O2 and . . .", ". . .continuous O2/N2 observation. . .", and ". . .the (O2/N2) ratio is. . ." are used on page 3 (line 15), page 4 (line 4), and page 7 (line 9), respectively. These expressions should be reconsidered throughout the manuscript. Similarly, the expressions of CO2 should also be reconsidered throughout the manuscript. For example, "CO2 mixing ratio" (e.g. page 8, line 18) and "CO2 concentration" (e.g. page 9, line 24) are used in the manuscript.*

In accordance with the reviewer's suggestion, we have reconsidered the expressions of $O_2$, $O_2/N_2$ ratio and so on throughout the manuscript. When those wordings have little distinctions in meaning, we have used $O_2$. We have also reconsidered the expressions of $CO_2$ throughout the manuscript. Since "mole fraction" is used in the explanation of $\delta(O_2/N_2)$ definition in Introduction, the expressions of "$CO_2$ mixing ratio" and "$CO_2$ concentration" have been changed to "$CO_2$ mole fraction".

Technical corrections:
*1) P2, L18: Change ":" after Naegler et al., 2007 to ";".*

"···Neagler et al., 2007: …" has been changed "…Neagler et al., 2007; …"

*2) P4–5, 2.1 Analytical system: Uniform the names of parts in the system in the text and Figure 1. For example, "glass vessel", "4-way 2-position valve", and "piezo actuator valve" are used in the text, but these are showed as "glass flask", "2-position valve", and "variable valve" in Figure 1.*

To respond the reviewer's suggestion, we have changed "Pump" to "Diaphragm pump", "Cooler" to "EPSC module", "2-position valve" to "4-way 2-position valve", and "Variable valve" to "Piezo actuator valve" in Figure 1. In addition, we have changed "mechanical mass flow controller" to "needle valve" in the manuscript to respond the comment of the reviewer #2. In accordance with this change, we have changed "Flow controller" to "Needle valve" in Figure 1.

*3) P8, L1: I think it would be better to add some words to make the readers focus to Figure 5. For example, "As shown in Fig. 5, ".*

"The observed $\delta(O_2/N_2)$ showed …" has been changed to "As shown in Fig. 5, the observed …".

*4) P10, L1 and 9: I think it would be better to switch the order of CO2 and (O2/N2).*

The order of $CO_2$ and $\delta(O_2/N_2)$ has been switched.

*5) P12, L5: Remove "- (hyphen)" from "the -variation".*

The hyphen has been removed.

*6) Units in section 2: Units of "cm3 min-1" and "cm3" are used as flow rate and volume in the text, but those in Figure 1 are "mL/min (or L/min)" and "L". Please uniform the units throughout the manuscript.*

The units of "mL min$^{-1}$" and "L min$^{-1}$" in Figure 1 have been changed to "cm$^3$ min$^{-1}$" and "$\times 10^3$ cm$^3$ min$^{-1}$", respectively.

*7) Figure 4 a: I think "Δ" in the label of vertical axis should be removed.*

We have redrawn Figure 4a as suggested.

*8) Figure 6 b: It is not clear the apparent variations of several tens of ppm amplitudes and 20s intervals in this figure. It would be better to add the expanded figure of apparent variations.*

We have added an inset in Figure 6b to show a closeup of the apparent variations.

*9) Figure 9: It would be very informative to add the cruise information in this figure. For example, changing the color depending on cruises, adding cruise-name labels. . .etc.*

We have added partition lines for individual cruises in the figures and cruise numbers at the top of figure.

*10) Figure A1: Modify from "Figure A12" to "Figure A1".*

In accordance with the suggestion of the reviewer #2, we have changed the appendix figure to normal figure. According to this change, the label of "Figure A1" has been changed to "Figure 11" and "Figure 11" of the original manuscript to "Figure 12" in the revised manuscript.

---

## Author Comment (AC2) · 8 Jun 2018

*In this manuscript, Hoshina et al. describe the construction, installation and performance of a fuel-cell/NDIR-based instrument for measuring O2 and CO2 aboard ships of opportunity. They also present a year of data collected while underway across the North Pacific. Overall, this is a very straightforward presentation of careful work. Nothing about this project is revolutionary or ground-breaking, but the continued development of O2-CO2 systems is important for the advancement of atmospheric potential oxygen (APO) as a useful oceanographic tool, and the data they have collected fill an significant gap in the APO community's records. This is valuable work and definitely deserves to be published. I have very few scientific questions or concerns about the content. With many years of excellent work in this field, the Tohjima lab knows what it is doing, and this paper reflects that expertise. It is also quite well written.*

We would like to thank you the anonymous referee for his/ her careful reading and for constructive feedbacks. We have revised the manuscript and describe the changes in the following. The referee's comments are in *blue italics*, our responses to the specific comments below.

Questions/comments on substance

*P3 line 3: Change to "The mole fraction is not used as a measure of O2 abundance because, for example, a change" Furthermore, it would be helpful if the authors gave an explanation of why mole fraction changes when per meg doesn't. This is alluded to on page 7 (around eq. 4), but it should be stated more explicitly at the outset.*

In accordance with the reviewers' comments, we have changed the relevant sentence to "The mole fraction is not used as a measure of $O_2$ abundance because the changes in the mole fraction of major atmospheric constituents like $O_2$ are sometimes very confusing. For example, adding 1 μmol of $O_2$ to an air parcel containing 1 mol of dry air results in a 0.79-ppm increase in the $O_2$ mole fraction and adding 1 μmol of $CO_2$ results in not only a 1-ppm increase in the $CO_2$ mole fraction but also a 0.21-ppm decrease in the $O_2$ mole fraction. These confusing results are attributed to influences of the changes in the total number of moles in the air parcel on the mole fractions or dilution effect (e.g., Keeling et al., 1998; Tohjima 2000). However, adding 1 μmol of $O_2$ to 1 mol of dry air, which contains 0.2094 mol of $O_2$ (Tohjima et al., 2005), always results in a 4.77-per meg change in the $\delta(O_2/N_2)$ value." (P3 line3-10).

*P5 line 5-6: I believe you are trying to say that the outlet pressures (and by extension, the pressures in the analysis cells), are kept at the same absolute value at all times by actively matching them to a reference volume. If this is what you do in fact mean to say, you should probably make it a bit more explicit.*

Yes, your understanding is correct. In accordance with your suggestion, we have changes the relevant sentence to "The outlet pressures of the analyzers are kept at the same absolute value at all times by actively matching them to a reference pressure using the piezo actuator valve and a differential pressure sensor (Model 204, Setra Systems, USA)." (P5 line 12-14).

*P5 line 11: Is the "mechanical mass flow controller" the same as the one mentioned on P4 line 25? If so please make it clearer. If not, specify this one more completely.*

The "mechanical mass flow controller" is not same as the "mass flow controller" on P4 line 25, but a kind of needle valve. To mention it clearly, we have changed the wording "a mechanical mass flow controller" to "a needle valve (2204, KOFLOC, Japan)" (P5 line 18). According to this change, we have also changed the labels "Flow controller" to "Needle valve" and changed the symbol of the valve in Fig. 1.

*P5 line 23: Is there more than one SUS tube? If so, explain how many and why.*

Yes, one SUS tube is used for the cold trap. To mention it clearly, we have changed the relevant part to "Figure 2a shows … two disk-shaped aluminum blocks, a 1/8-inch SUS tube, and a drum-shaped glass vessel with a volume of about $1.0 \times 10^3$ cm$^3$. The aluminum blocks had four grooves for the 1/8-inch SUS tubes, including spare tubes to address clogging." (P5 line 28-P6 line 3).

*P6 line 10: To what do you attribute the improved performance of the 2nd trap design? Is it simply larger glass volume, more complete chilling of the glass due to the insulation, the coaxial design of the glass trap. . .? Please give a few words of explanation.*

We change the trap design for more complete chilling. To clarify it, we have changed the relevant sentence to "due to the more complete chilling of the glass vessel" to the end of the relevant. (P6 lines 16-17).

*P6 line 17: At what interval did you operate your changeover valve? "1-5 minutes" is*

*not specific enough, as the frequency of these switches can have a significant impact on the precision of your results (see, for example, Keeling et al, 2004). How did you settle on this interval?*

To respond the reviewers' comments, we have changed the ambiguous descriptions to "In previous studies (e.g., Stephens et al., 2007; Thompson et al., 2007; van der Laan-Luijkx et al., 2010; Goto et al., 2013), the sample and reference air are alternately introduced into each fuel cell sensor by switching the 4-way 2-position valve at 1- to 5-min intervals. In this study, we adopted 2 min for the valve-switching intervals in light of the responses of the $O_2$ and $CO_2$ analyzer after valve switching, as described below." (P6 lines 22-25).

*P6 line 20-21: It's not at all clear from the plot that the CO2 ever reaches a plateau before the changeover valve changes state again. My concern is that the CO2 values you infer will be consistently biased toward the value of whatever gas was previously in the analysis chamber. Perhaps a better approach is to fit each transition with function like (1 – exponential), with 2 free parameters (rise time and asymptote) and use the asymptote as the true mixing ratio, even this value is never measured in the instrument.*

As the reviewer suggested, Figure 3 didn't clearly show whether the $CO_2$ signal reaches a plateau during the 2-min intervals. To respond the reviewer's concern, we have redrawn Figure 3 to show more details of the changes in the $CO_2$ signals. We hope the added Figure 3d convince the readers that the $CO_2$ signal reaches a plateau after 90 s after the valve switching. In the original manuscript, we mentioned that the last 10 s the signals were averaged to calculate $\Delta CO_2$, but actually we used the last 20 s data. Consequently, the sentences "The signal plateau … last 10 s of each 2-min interval" (P6 line20-P7 line3) have been changed to "The signal plateaus at least 1 min after the valve switching, and the output signal is averaged from the second minute of the cycle (Fig. 3c). The deviation of the $O_2$ mole fraction in the sample gas from that of the working reference gas for the $i$-th 2-min interval, $\Delta O_{2,i}$, is computed based on the 1-min average according to the following equation:

$$\Delta O_{2,i} = (-1)^{(i-1)} \left[ v_i - (v_{i-1} + v_{i+1})/2 \right]/2 \qquad (1)$$

where $v_i$ represents the average of the output signal for the second minute of the $i$-th 2-min interval and the output signal represents the difference of the sample gas minus working reference gas when $i$ is an odd number greater than 1.

In contrast to the $O_2$ analyzer, the $CO_2$ analyzer alternately measures the sample and

working reference gases. The temporal variation of the output signal of the $CO_2$ analyzer is depicted in Figs. 3d and 3d. As shown in the figures, the output signal does not plateau until after more than 90 s because of the relatively low flow rate in comparison with the volume of the optical cell of the LI-840A analyzer (14.5 $cm^3$). Therefore, we compute an average of the output signal for the last 20 s of each 2-min interval." (P6 lines 29-P7 line 12).

*P6 lines 26: What happens when i=1? I recognize that this formula represents the difference between a block and the average of the blocks before and after, but the formula doesn't work for the first block after a calibration run.*

When i=1, we did not calculate for $\Delta O_2$ and $\Delta CO_2$. We have added "greater than 1" in the manuscript. (P7 line 6-7 and line 16).

*P7 line 24-25: I'm a bit confused by your calibration procedure. I think you're saying that you used two span tanks for O2 and calculated a linear response function for the Oxzilla every 25 hours. How did you interpolate the response of the instrument for the many observations that were made during the intervening 23.5 hours? Similarly, it seems like you had a 3-point calibration for CO2 made just once at the beginning of the week of observations. If you have a 3-point calibration, did you assume a linear response function, or did you allow a 3-parameter form with some curvature? Am I correct that you did not repeat the CO2 calibration at any time during or after the 7-day period of observations? Please clarify these points.*

We used linear response functions both for the $O_2$ and $CO_2$ analyzers. And we determined a single calibration line for the $O_2$ values from the repeated measurements of the two $O_2$ reference gases during the 7-day observations. We didn't repeat the $CO_2$ calibration procedure during and after the preliminary measurements. To mention above clearly, we have changed the sentences to "We determined a single calibration line of linear response function for $\delta(O_2/N_2)$ values from all the measurements of the two standard gases during the observation. As for the $CO_2$ mole fraction, a single calibration line of linear function was determined from measurements of three standard gases with 387, 406, and 434 ppm only before the observation." (P8 line 6-9).

*P8 line 19: I think you mean "During every 24hr period, these three reference gases were measured for 32min each." If that's right, please reword. If not, clarify.*

Your understanding is correct. We have changed to "During every 24 h period, these three standard gases were measured for 32 min each." (P9 line 4).

*P8 line 28: There is no way for us to tell (from Fig. 6b) that there are 20s cycles. It would be very helpful if there were an inset figure that zoomed in on a subset of the data with a much-expanded time-scale.*

To respond the reviewer's comment, we have added an inset figure in Fig. 6b, which shows closeup variations for 60 seconds period.

*P9 line 15-16: Are the "hourly averages" mentioned on line 16 tank values, or atmospheric data? If the latter, wouldn't natural variability in the atmosphere lead you to expect more scatter? See also my comments on Fig. 7 below.*

The "hourly averages" are tank values. We have changed "the expected standard deviation of hourly average of the $\delta(O_2/N_2)$ value" to "the expected standard deviation of the hourly $\delta(O_2/N_2)$ value for the standard gases" (P10 line 2).

*P10 lines 10-14: I am puzzled about why the authors didn't just deconvolve the flask errors to get a separate value for the uncertainty in the shipboard data. If the (shipboard data – flask values) have a scatter of XX per meg and the flasks themselves have an uncertainty of YY, then isn't the uncertainty in shipboard data alone just given by $ZZ = sqrt(XX^2 – YY^2)$? This way one can report a meaningful error for the system, rather than an upper limit.*

In accordance with the reviewer's suggestion, we have reported the estimated uncertainty not the upper limit. Additionally, since we have updated the $\delta(O_2/N_2)$ values of the flask samples in association with an update of a working reference gas used for the flask measurements in our laboratory, which was a normal process, we have recalculated the shipboard-flask differences and redrawn Fig. 10. Consequently, the relevant two sentences have been changed to "The averaged differences with standard deviations were $-2.8\pm9.4$ per meg of $\delta(O_2/N_2)$, $-0.02\pm0.33$ ppm of $CO_2$, and $-2.9\pm9.5$ per meg of APO. Taking into account the uncertainties of the flask measurements (5 per meg for $\delta(O_2/N_2)$ and 0.05 ppm for $CO_2$ measurements, Tohjima et al., 2003), we conclude that the uncertainties of the in situ measurements aboard NC2 were 8.0 per meg for $\delta(O_2/N_2)$ and

0.33 ppm for $CO_2$." (P10 line 27-P11 line 4).

*P10 line 20: I cannot see the "noticeable decrease" in Figure 9. Perhaps a zoomed-in inset would show it. In fact, when I look at Figure 8, I think I seen an _increase_ in APO during the eastbound leg of NC2-125.*

We are afraid that our ambiguous description, probably wordings of "decrease" and "increase", confused the reviewer. Here, we mentioned the low APO values ($< -180$ per meg) during NC2-125 and high APO values ($> -100$ per meg) during NC2-127. To mention them clearly, we have changed the sentence, "For example, the APO values show … in late May", to "For example, the APO values show relatively low values ($< -180$ per meg) during the eastbound voyage NC2-125 in early March and high values ($> -100$ per meg) during the eastbound voyage NC2-127 in late May." (P11 line 9-11).

*P11 line 5: The period 2014-2016 is not 4 years in length. Please correct either the dates or the duration.*
"2-yr" is correct. We have changed "during the 4-yr period 2014-2016" to "during the 2-yr period 2014-2016" (P11 line 24).

*P11 line 7: Presumably the figure referenced here is the one labelled "A12". It's odd to me that this one figure would comprise an entire appendix. I would suggest simply making it a part of the normal figures (since it is a valuable one). At the very least, make sure it is correctly referenced in the text.*

In accordance with the reviewer's suggestion, we have changed the appendix figures (Fig. A1) to the normal figures, which are referred to as Figure 11, and "Figure 11" of the original manuscript to "Figure 12" in the revised manuscript.

*Figure 3: This would be much more valuable if there were fewer cycles shown with more detail. Perhaps an inset with one representative cycle (for each of plots "a" and "b") would make the true stability of the measurements more apparent.*

As is mentioned in the responses to the reviewer's comments (P6 line 20-21), we have redrawn Figure 3 to show fewer cycles and detail variations near the end of the individual cycles.

*Figure 7: I am puzzled that the error bars on the black circles (when the ship is in port) are not apparently any smaller than the scatter in the blue points (taken at sea). I'm just going by eye, but it certainly looks like the error bars in port capture more than 68% of the blue points. Wouldn't the rocking of the ship and the resulting response in the Oxzilla II make the scatter a bit larger when at sea? In lines 19-20 you say the scatter is actually bigger in the blue points, but I just can't see that. Also, on lines 20-21, you say you are only using the in-port measurements for calibration of the Oxzilla. Really? What if the instrument response varies during the time between ports? Why not use the information from the tank runs taken at sea (the blue points in Figure 7) to address this possibility.*

We are afraid that the rather large error bars on the black circles, which sometimes exceed 10 per meg, confuse the reviewer. Unfortunately, the Oxzilla-II at port was not so stable as it was in our laboratory. However, we continued the measurements of the standard gases at port for more than 5 hours for the individual standard gases. So, the standard errors were smaller than 2 per meg. Therefore, we calibrated the instrument by using the calibration lines based on the standard gas measurements before and after each round-trip voyage as is mentioned in the manuscript. Consequently, we have changed the last four sentences in Section 3.1 to "In Fig. 7, the average $\delta(O_2/N_2)$ values of the standard gases determined when NC2 was berthed at the port of Tahara are also plotted as black circles with error bars showing the standard deviations. Unfortunately, the standard deviations for the standard gases were larger than the expected values obtained in our laboratory, as discussed in Section 2.3. However, the standard errors were lower than 2 per meg because the measurements were continued for more than 5 h for the individual standard gases. Therefore, we calibrated the O2 analyzer using calibration lines based on the results at the port just before and after each round-trip voyage." (P10 line 3-9).

Strictly editorial comments:

In accordance with the reviewer's comments, the manuscript has been changed except the following comments:

*P3 line 17: Change to "trasects of the data in the western Pacific region revealed that variation in the magnitude of the bulge in annual mean APO"*

We have changed to "transects of the data in the western Pacific region revealed that

variation in the magnitude of the bulge in annual mean APO".

*P6 line 25: Change to "signal for the second minute of the i-th 2-minute interval"*

We have changed to "signal for the second minute of the $i$-th 2-min interval".

*P7 line 6: Change to "the average of the last 10s of data for the i-th 2-minute interval."*

We have changed to "the average of the last 20 s of data for the i-th 2-min interval". Because we mistook the averaged span, "20 s" is corrected.

*P8 line 1: Change to "O2/N2) shown in Figure 5, reveals a diurnal cycle"*

In accordance with the comments from both reviewers #1 and #2, "The observed $\delta(O_2/N_2)$ showed a diurnal cycle …" has been changed to "As shown in Fig. 5, the observed $\delta(O_2/N_2)$ revealed a diurnal cycle ...".

*P8 line 16: Change to "-579 mer peg (tank #CPD-00010)"*

"(CPD-00010)" has been changed to "(tank #CPD-00010)", and "(CPD-00011)" has also changed to "(tank #CPD-00011)", and (CPB-17350)" has also changed to "(tank #CPB-17350)".

*P8 line 19: Change to "During the shipboard measurements"*

As the above your suggestion, we changed to "During every 24 h period, these".